

# Comprehensive evaluation of drought stress on medicinal plants: a meta-analysis

Uğur Tan and Hatice Kübra Gören

Field Crops, Aydın Adnan Menderes University, Aydın, Türkiye

## ABSTRACT

Drought stress significantly affects plants by altering their physiological and biochemical processes, which can severely limit their growth and development. Similarly, drought has severe negative effects on medicinal plants, which are essential for healthcare. The effects are particularly significant in areas that rely mostly on traditional medicine, which might potentially jeopardize both global health and local economies. Understanding effects of droughts on medicinal plants is essential for developing strategies to enhance plant adaptability to drought stress, which is vital for sustaining agricultural productivity under changing climatic conditions. In this study, a meta-analysis was conducted on 27 studies examining various parameters such as plant yield, chlorophyll content, relative water content, essential oil content, essential oil yield, non-enzymatic antioxidants, enzymatic antioxidants, phenols, flavonoids, and proline content. The analysis explored the effects of drought across different stress conditions (control, moderate, and severe) to gain deeper insights into the drought's impact. The categorization of these stress conditions was based on field or soil capacity: control (100–80%), moderate (80–50%), and severe (below 50%). This classification was guided by the authors' descriptions in their studies. According to meta-analysis results, enzymatic antioxidants emerge as the most responsive parameters to stress. Other parameters such as relative water content (RWC) and yield also exhibit considerable negative mean effect sizes under all three stress conditions. Therefore, when evaluating the impacts of drought stress on medicinal plants, it is beneficial to include these three parameters (enzymatic antioxidants, RWC, and yield) in an evaluation of drought stress. The chlorophyll content has been determined not to be a reliable indicator for measuring impact of drought stress. Also, measuring antioxidants such as flavonoids and phenols could be a better option than using radical scavenging methods like DPPH (2, 2-difenil-1-pikrilhidrazil), FRAP (ferric reducing antioxidant power), and ABTS (2, 2′-Azino-bis (3-ethylbenzothiazoline-6-sulfonic acid)).

Corresponding author
Uğur Tan, ugur.tan@adu.edu.tr

## INTRODUCTION

How bad could climate change get? As early as 1988, the landmark Toronto Conference declaration described the ultimate consequences of climate change as potentially "second only to a global nuclear war." Despite such dramatic proclamations decades ago, the

climate catastrophe remains relatively understudied and poorly understood (*Kemp et al., 2022*).

Among the factors affected by climate change, temperature increases, elevated carbon dioxide levels, glacial runoff, changes in precipitation regimes, and interaction of these factors affect plant growth and development (*Webster, 2008*; *Gornall et al., 2010*). For instance, climatic conditions such as temperature, precipitation and extreme occurrences like droughts, floods and wind storms/tropical cyclones impact crop yields. Also, beyond a specific temperature range, warming reduces yields due to crops growing faster, resulting in less production (*Asseng, Foster & Turner, 2011*). Climate change significantly impacts crop and livestock production. Annual yield losses of 40 million tons or US $5 billion have been projected due to global warming since 1981 (*Lobell & Field, 2007*). Given that 75% of the world's population is depends on agriculture, which provides a living for 2/3 of people, and the fact that the world's six main crops (rice, wheat, corn, sorghum, soybeans and barley) occupy 40% of cropland, contribute 55% of non-meat calories, and account for more than 70% of animal feed; any impact on these crops will have a detrimental effect on global food security (*Alam & Satpati, 2021*).

Similarly, drought has severe adverse effects on medicinal plants, which are crucial for healthcare and hold significant economic importance. These impacts are especially pronounced in regions heavily dependent on traditional medicine, posing potential threats to global health and local economies (*Selmar et al., 2017*). A key distinguishing feature of medicinal plants is their richness in secondary metabolites such as alkaloids, terpenoids, and phenolics, which possess therapeutic properties. These compounds are less common or absent in many non-medicinal plants (*Alamgir, 2017*).

Drought stress reduces plant biomass, which is particularly critical for medicinal plants as it directly impacts the availability of raw materials for drug production (*Farooq et al., 2011*). Additionally, drought stress significantly alters the physiological and biochemical processes in plants, severely limiting their growth and productivity. This further exacerbates the challenges faced by medicinal plants in maintaining their therapeutic efficacy (*Seleiman et al., 2021*).

Drought stress significantly affects plants by altering their physiological and biochemical processes, severely limiting growth and development (*Seleiman et al., 2021*). Exposure to drought leads to decreased water potential and turgor pressure within cells, inhibiting growth in roots, stems, leaves, and fruits (*Bashir et al., 2022*). To adapt these conditions, plants modify their metabolic pathways and biochemical activities to adapt better to these conditions (*Hemati et al., 2022*). Also, drought can disrupt cellular homeostasis and consequently reduce crop yields (*Hemati et al., 2022*). Additionally, drought stress impairs plant-water interactions, decreasing water-use efficiency (*Farooq et al., 2009*).

When plants exposed water scarcity (drought stress), they often respond by closing their stomata. Stomata are small pores on the surface of leaves that regulate gas exchange, including water vapor loss through transpiration. Closing stomata helps conserve water but can also limit carbon dioxide uptake for photosynthesis. Drought stress can lead to the generation of reactive oxygen species (ROS), such as hydrogen peroxide and superoxide

radicals, which can damage cellular components (*Mittler et al., 2004*). Plants exhibit diverse responses to drought stress, encompassing morphological, physiological, and biochemical changes. These adaptations are critical for survival under water-limited conditions. Morphologically, plants may alter leaf size to minimize water loss (*Gupta, Rico-Medina & Caño-Delgado, 2020*). Physiologically, adjustments include stomatal closure to conserve water and modifications in photosynthetic activity to adapt to the reduced water availability, significantly impacting processes like $CO_2$ assimilation and energy production (*Reddy, Chaitanya & Vivekanandan, 2004*). Biochemically, plants increase the synthesis of antioxidant enzymes to prevents oxidative stress induced by drought (*Takahashi et al., 2020*). Understanding these responses is essential for developing strategies to enhance plant adaptability to drought stress, which is vital for sustaining agricultural productivity under changing climatic conditions.

Understanding the mechanisms which plants manage water stress is essential for agronomists and plant breeders to develop crops that are more resilient to drought conditions. This understanding facilitates the development of varieties that can maintain stable yields in regions prone to water scarcity, thereby ensuring food security (*Gupta, Rico-Medina & Caño-Delgado, 2020*). Also, knowledge of the specific genes and biochemical pathways involved in drought tolerance is crucial for genetic engineering and selective breeding techniques. This allows for the development of plant varieties that are specifically engineered to endure dry conditions without reducing crop quality or yield (*Ashraf, 2010*; *Takahashi et al., 2020*).

Meta-analysis, essentially an 'analysis of analyses,' serves as a quantitative method that synthesizes findings from multiple independent studies, enabling comprehensive overviews (*Glass, 1976*). This approach helps in identifying gaps within existing research, thereby suggesting new directions for future studies (*Kuznetsov, Passot & Sulem, 2008*). By facilitating the statistical examination of mean effect sizes, meta-analysis offers an objective means to assess and compare the experimental results of different researchers (*Mak-Mensah et al., 2021*). Moreover, it enhances the capacity to comprehensively evaluate theories and compare the efficacy of treatments across diverse settings (*Luo, Wang & Sun, 2010*). It is vital for integrating data across different studies, allowing for a more comprehensive understanding of drought tolerance mechanisms. It helps identify important agronomic, physiological and biochemical properties involved in drought response across various crops, consequently improving the accuracy of the conclusions drawn from individual studies (*Kumar et al., 2020*). Meta-analysis quantifies variability and consistency in how different species or genotypes respond to drought, providing insights into the most effective traits and treatments for enhancing drought tolerance. This approach allows researchers to pinpoint which interventions are most likely to be successful across different environments and plant species (*Bartlett, Scoffoni & Sack, 2012*).

This study primarily focuses on evaluating the impact of drought on medicinal plants, which are known for their high secondary metabolite production, including essential oil content, essential oil yield, and non-enzymatic antioxidants. The aim is to analyze how drought affects these specific parameters and to compare them with commonly studied parameters (such as enzymatic antioxidants, proline, relative water content *etc.*) that are

used in general plant stress research. This comparison helps assess their relative effectiveness in responding to drought stress. Medicinal plants are specifically chosen as the study population.

There are two main reasons for using medicinal plants in this study. Firstly, studies suggest that the medicinal properties of plants can increase under stress. According to *Selmar & Kleinwächter (2013)* and *Omidi et al. (2018)*, drought stress can lead to a higher concentration of secondary metabolites such as alkaloids, phenols, and terpenoids, which are responsible for the therapeutic effects of medicinal plants. Drought-induced stress activates metabolic pathways that enhance the production of these metabolites, potentially increasing the medicinal value of the plants. Specifically, drought stress has been found to stimulate the production of specialized metabolites like terpenoids and triterpenoids, which are beneficial for the medicinal properties of these plants (*Yang et al., 2022*).

Secondly, some studies suggest that medicinal plants possess high bioactive and antioxidant properties, making them more tolerant to stress conditions. These metabolites contribute to the plant's resilience under stress, further enhancing their medicinal efficacy.

Under drought stress, plant tissues activate protective mechanisms against the detrimental effects of ROS. This includes an increase in the activities of antioxidant enzymes and higher concentrations of various antioxidant secondary metabolites (*Sharma & Dubey, 2005*).

In this study, we performed a meta-analysis to (1) determine drought stress responses of medicinal plants, (2) understand the relationship between parameters and drought stress level and (3) compare the effectiveness of these responses which parameters are more responsive to drought stress.

Meta-analysis will be extremely useful in our study, improving our understanding of how plants respond to drought by combining data from many studies. This method provides a broader perspective on the mechanisms that allow plants to tolerate drought. By pinpointing specific physiological and biochemical responses to drought, scientists and researchers can create focused strategies to mitigate stress-related damage.

## MATERIALS AND METHODS

This meta-analysis study was prepared according to the Preferred Reporting Items for Systematic Reviews and Meta-analysis (PRISMA) 2020 statement.

### Bibliographic research and data collection

During the last quarter of 2023 and the first quarter of 2024, variety keyword (mentioned below) searches were conducted using scientific bibliographic platforms such as Google Scholar and Web of Science to explore the effects of drought stress on medicinal plants. We also examined the cited references in relevant review articles. Initially, our searches yielded 867 records: 321 from Web of Science and 546 from Google Scholar. After removing duplicates (156 records) and records deemed ineligible by automation tools (293 records), as well as others excluded for various reasons such as missing data, SD or SE values (332 records), 86 records were screened. Of these, 17 were further excluded, leaving 69 reports for retrieval. Ultimately, 67 reports were assessed for eligibility, but only

PRISMA 2020 flow diagram for new systematic reviews which included searches of databases and registers only

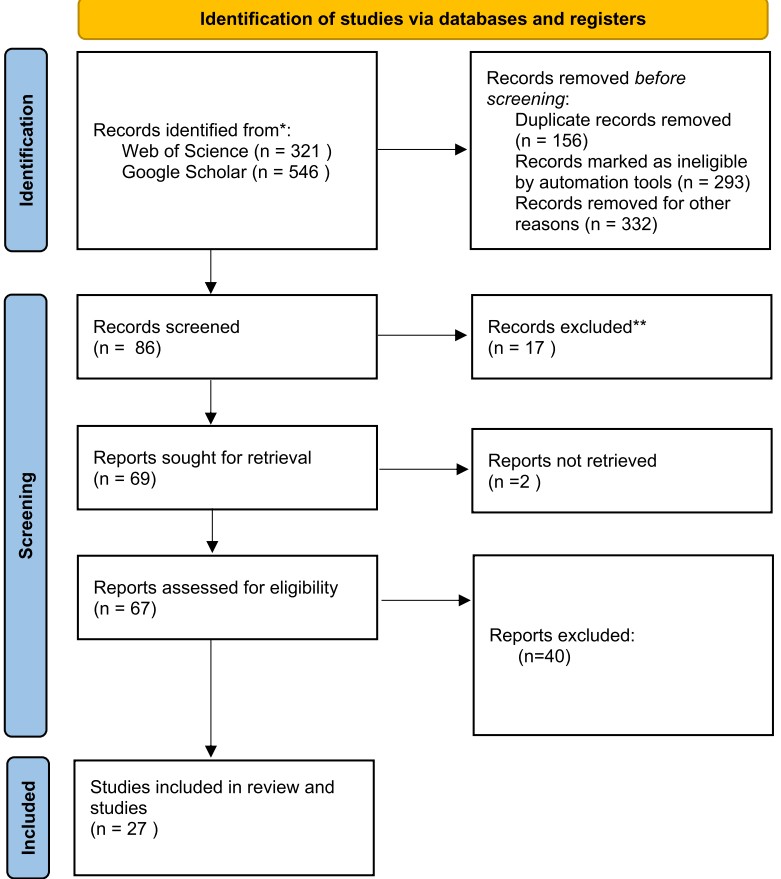

**Figure 1 Preferred reporting items for systematic reviews and meta-analyses (PRISMA) flow diagram for the meta-analysis.** From: *Page et al. (2021)*.

27 studies met the criteria and were included in the review. Additionally, two reports could not be retrieved, as depicted in flow diagram Fig. 1.

Different stress conditions (control, moderate, and severe stress) were employed to evaluate the effects of drought. The categorization of treatments into control, moderate and severe was based on the authors' descriptions. Consequently, stress conditions were defined as follows: Control or no stress (100–80% field or soil capacity), moderate (80–50% field or soil capacity), and severe (below 50% field or soil capacity).

The keywords "drought," "water deficit," "water treatments," "water levels", "irrigation levels", "irrigation" and "water scarcity" were used in combination with terms such as "medicinal plants," "essential oil," "essential oil content," "essential oil yields," and "antioxidants." Additionally, the Latin names of some well-studied medicinal plants were included to refine the search. Some parameters (enzymatic antioxidants, proline, relative water content) that have been less frequently measured were also specifically targeted in our search strategy. Additionally, the references of studies were reviewed to identify suitable studies for inclusion in the meta-analysis.

**Table 1 Comparative analysis of trait responses under three levels of drought stress (CM, CS, MS).**

| Traits | CM | | | | | CS | | | | | MS | | | | |
|---|---|---|---|---|---|---|---|---|---|---|---|---|---|---|---|
| | Effect size | LL | UL | n | p-value | Effect size | LL | UL | n | p-value | Effect size | LL | UL | n | p-value |
| YD | −1.356 | −1.732 | −0.979 | 61 | <0.000*** | −5.089 | −5.956 | −4.223 | 38 | <0.000*** | −2.704 | −3.246 | −2.162 | 34 | <0.000*** |
| Chl | 0.436 | −0.138 | 1.009 | 20 | 0.137ns | −0.752 | −1.492 | −0.012 | 24 | 0.047* | −0.117 | −0.695 | 0.461 | 19 | 0.691ns |
| EOC | 0.641 | 0.206 | 1.077 | 35 | 0.004*** | 0.515 | −0.290 | 1.320 | 23 | 0.210 | −1.137 | −3.980 | −0.395 | 21 | 0.003*** |
| EOY | −0.152 | −0.982 | 0.677 | 27 | 0.719ns | −2.899 | −4.454 | −1.344 | 15 | <0.000*** | −3.207 | −4.810 | −1.605 | 15 | <0.000*** |
| Pro. | 1.817 | 0.710 | 2.924 | 9 | <0.000*** | 1.352 | 0.411 | 2.293 | 8 | 0.005*** | 0.441 | −0.377 | 1.259 | 7 | 0.291ns |
| AO | −0.091 | −0.696 | 0.514 | 26 | 0.768ns | −0.226 | −1.183 | 0.732 | 20 | 0.644ns | −0.451 | −1.157 | 0.255 | 20 | 0.211ns |
| EAO | 4.017 | 1.184 | 6.849 | 15 | 0.005*** | 12.248 | 7.834 | 16.662 | 13 | <0.000*** | 10.962 | 6.730 | 15.193 | 11 | <0.000*** |
| RWC | −1.329 | −1.927 | −0.730 | 18 | <0.000*** | −3.226 | −4.165 | −2.286 | 11 | <0.000*** | −1.471 | −2.032 | −0.910 | 10 | <0.000*** |
| Phn | 0.958 | 0.299 | 1.616 | 23 | 0.004*** | 1.612 | 0.791 | 2.432 | 21 | <0.000*** | −0.343 | −1.107 | 0.421 | 17 | 0.379ns |
| FLV | 2.277 | 0.932 | 3.623 | 9 | 0.001*** | 2.060 | 1.118 | 3.002 | 13 | <0.000*** | −0.308 | −1.453 | 0.837 | 9 | 0.598ns |

**Note:**
Effect sizes, confidence intervals, and statistical significance. LL, Lower Limit; UL, Upper Limit; N, Number of samples; YD, Yield; Chl, Chlorophyll Content; EOC, Essansial oil content; EOY, Essential oil yield; Pro, Proline Content; AO, Enzymatic Antioxidants; EAO, Non- Enzymatic Antioxidants; RWC, Relative Water Content; Phn, Phenole; FLV, Flavonoid; CM, Control vs. Moderate Stress; CS, Control vs. Severe Stress; MS, Moderate vs. Severe Stress. *: $p < 0.05$; significant at the 5% level, ***: $p < 0.001$; highly significant at the 0.1% level.

The literature search was conducted across several databases, divided among the authors (Dr. Uğur TAN and Dr. Hatice Kübra GÖREN) to ensure thorough coverage. Each author independently scanned the literature and selected appropriate studies. The findings were then collectively reviewed, compared, and evaluated. Studies that were unsuitable were excluded, ensuring that only the most suitable studies were included in our analysis (criteria stated at below). References and some information about references included to meta-analyses were presented in Table 1.

The observations analyzed from these studies are as follows; yield (fresh or/and dried), chlorophyll content (SPAD or/and chlorophyll a/b), essential oil content, essential oil yield, proline, non-enzymatic antioxidant (DPPH, FRAP, ABTS), enzymatic antioxidants (POD, CAT, SOD), relative water content, phenol and flavonoid.

The meta-analysis was conducted using data from published, peer-reviewed studies that need a range of specific criteria; however, not all criteria needed to be met for inclusion. To isolate the effects of drought stress, studies have additional applications such as fertilizers or hormones were excluded and only control ones included to the analysis. This is because the application may have a biased effect on certain parameters in response to drought stress. Both greenhouse and field studies were considered adequate for inclusion, ensuring a broad expression of conditions. Heterogeneity results were given in results section for each parameter. Subgroup analysis was not employed in this study because a sensitive meta-analysis requires at least 10 studies per parameter to ensure reliable results. However, employing subgroups would reduce this number, potentially compromising the reliability of the findings. The study eligibility criteria for inclusion in this meta-analysis were carefully designed to select studies that meet specific parameters essential for conducting a robust analysis of medicinal and aromatic plants under drought stress conditions. The criteria are as follows:

1) *Availability of quantitative data*: Studies must provide essential quantitative data, including absolute mean values, standard deviation, standard error, and sample size. These data parameters are crucial for conducting meta-analysis and resulting meaningful conclusions.

2) *Quantification of drought stress*: Drought stress conditions must be clearly expressed as words (control, moderate, and severe) and numerically, typically as a percentage reduction in water availability. This ensures consistency and comparability of drought treatments across different studies.

3) *Plant species selection:* Studies must involve species known to have essential oil content, essential oil yield from medicinal and aromatic plants. This criterion focuses the review on a specific category of plants relevant to the study's objectives.

4) *Control group specification*: Studies included in the analysis should have an untreated control group. This criterion is particularly important when assessing the impact of drought stress exclude other treatments (*e.g.*, fertilizers, bacteria, nano-fertilizers) to isolate and evaluate the specific effects of drought.

5) *Language and study design*: Only controlled studies published in English were considered eligible for inclusion. This language requirement ensures uniformity in data extraction and analysis.

These criterions were cautiously applied to ensure the selection of high-quality studies that meet standardized reporting and experimental conditions. By focusing on studies that meet these criterions, the meta-analysis aims to provide a comprehensive and precision synthesis of research findings related to the effects of drought stress on medicinal and aromatic plants. This approach enhances the reliability and applicability of the meta-analysis results, contributing valuable insights to the field of plant stress physiology and agriculture.

## Sensitivity analysis

To assess whether any individual study equally influence the meta-analysis results, we conducted a 'one study removed' sensitivity analysis. This involved sequentially removing each study and observing the effects on the overall effect sizes. The analysis revealed that the exclusion of any single study did not lead to significant changes in the outcomes, indicating that no single study dominated the findings. Additionally, we evaluated the effect size weights of each study to further gauge sensitivity. It was determined that the studies contributed relatively similar weights to the meta-analysis results for each parameter.

## Statistical analysis

Mean and standard deviation values, sample sizes and standard error were used to obtain the meta-analysis results of the studies. These data were independently reviewed by the researchers. The effect size (Cohens' d and 95% confidence intervals [CI]), was calculated. Cohen's d is an effect size measure that quantifies the standardized difference between two means. Traditionally, it is categorized as weak (0–0.2), moderate (0.2–0.5), high (0.5–0.8),
and very high (above 0.8) (*Rosenthal, 1996*; *Cohen, 2013*). However, for the purposes of our specific study, we have modified these categories to better align with our observations and analytical framework. In our revised scale, an effect size from 0 to 1 is considered low, from 1 to 2 is medium, from 2 to 4 high and any value above four is classified as very high. This adjustment allows for a more accurate representation of the effect sizes relevant to our research context.

Additionally, for publication bias, the funnel plot asymmetric were determined and this visual impression is confirmed by Egger's test which yields a statistically significant *p*-value. If there is a publication bias "duwal and tweedie trim and fill" was used to re-computing the effect size at each parameter until the funnel plot is symmetric with new effect size. In theory, this will yield an unbiased estimate of the effect sizes. While this trimming yields the adjusted effect size, it also reduces the variance of the effects, yielding a narrow confidence interval. Therefore, the algorithm then adds the original studies back into the analysis, and imputes a mirror image for each. This fill has no impact on the point estimate but serves to correct the variance (*Duval & Tweedie, 2000a*, *2000b*).

Publication bias can be a common issue in research due to certain parameters not being normally distributed, as many studies lack the necessary data required for inclusion in meta-analysis. Consequently, we were unable to include every study in the meta-analysis. This limitation highlights the challenge of synthesizing findings across studies and underscores the importance of considering publication bias when interpreting results. Efforts to address this bias include acknowledging the potential impact of unpublished or selectively published studies on the overall conclusions drawn from meta-analytic studies. Comprehensive Meta-Analysis (CMA) Version 4 software was used to analyze meta-analysis data (*Borenstein et al., 2010*).

## RESULTS

The findings are categorized into three comparative scenarios: control *vs.* moderate stress, control *vs.* severe stress and moderate *vs.* severe stress. Each comparison not only shows the observed changes but also their statistical significance, effectively shows the varying responses of plants under different stress conditions.

*Control vs. moderate stress:* The analysis indicates significant alterations under moderate stress compared to control conditions across various parameters, such as proline content, antioxidant levels, and relative water content with the specifics of the changes (*e.g.*, increase or decrease) and their statistical metrics (mean effect sizes, confidence intervals, and *p*-values) reported.

*Control vs. severe stress:* This comparison generally shows more pronounced effects than moderate stress, with severe stress significantly impacting parameters like chlorophyll content, yield and antioxidant production more drastically. The magnitude and statistical significance of these changes are clearly documented.

*Moderate vs. severe stress:* The differences between these stress levels are also evaluated, it shows a trend where severe stress exacerbates the effects observed under moderate stress. Significant differences are reported and providing insights into the incremental impact of increasing stress severity.

*Heterogeneity and prediction intervals*: The results also include an evaluation of heterogeneity observed across studies, indicating varying effects of drought on different parameters and experimental conditions. Prediction intervals are provided for each parameter under different stress conditions, emphasizing the range of possible true effects and the substantial variability in plant responses to drought.

These results not only show the specific impacts of varying drought stress conditions but also establish a robust statistical foundation for further discussions and analyses within the ongoing academic studies on plant physiology under environmental stressors. A comparative analysis of different parameters responses to three levels of drought stress (CM, CS, MS) is including effect sizes, confidence intervals, and statistical significance, with comprehensive results presented in Table 2.

## Plant yield or plant biomass

The impact of drought stress on medicinal plant yield was significantly and negatively affected in all different stress levels. Under moderate stress conditions, medicinal plantyields significantly decreased with a mean effect size of −1.356 (95% CI [−1.732 to −0.979], and a Z-score of −7.059 ($p < 0.001$). Severe stress conditions further accelerated the reduction in yield, with a mean effect size of −5.089 (95% CI [−5.956 to −4.223] and an even more expressed Z-score of −11.514 ($p < 0.001$). Comparing moderate to severe stress, yields also decrease significantly, with a mean effect size of −2.704 (95% CI [−3.246 to −2.162] and a Z-score of −9.779 ($p < 0.001$) (Fig. 2A).

The analysis revealed high heterogeneity ($I^2$ = 69% to 81%, $p < 0.001$), indicating diverse responses to drought across different studies. The prediction intervals ranged broadly from −3.610 to −0.898 under moderate stress and from −9.356 to −0.823 under severe stress, underscoring the considerable impact of drought on yield variability and highlighting the complex, significant effects of stress levels on crop production.

## Chlorophyll content

The statistical analysis compared the effects of different stress levels on chlorophyll content in plants. When we examine control *vs.* moderate stress, the difference in mean effect size was found 0.436, with a 95% confidence interval (CI) ranging from −0.138 to 1.009. This variation was not statistically significant, as indicated by a Z-score of 1.489 and a *p*-value of 0.137, it means that moderate stress does not significantly alter chlorophyll content compared to control. Under control *vs.* severe stress, there was a significant decrease in chlorophyll content, with a mean effect size of −0.752 and a 95% CI of −1.492 to −0.012. The Z-score was −1.991, and statistical significance with a *p*-value of 0.047, highlighting that severe stress substantially reduces chlorophyll content. Lastly, moderate stress compares severe stress, comparison showed no significant difference, with a mean effect size of −0.117 and a 95% CI of [−0.695 to 0.461]. The Z-score was −0.397 and the *p*-value was 0.691, indicating that the impact of severe stress was not significantly different from moderate stress on chlorophyll content (Fig. 2B).

The heterogeneity among the studies was significant, with an $I^2$ of 68% and a Q-value of 60.155 ($p < 0.001$), suggesting considerable variation in how drought impacts

**Table 2 Summary and description of pooled studies.** Authors, plant species, investigation parameters and study conditions (field or greenhouse).

| Ref. no | References | Plant species | Parameters | Repetition | Study condition |
|---|---|---|---|---|---|
| 1 | *Chrysargyris et al. (2016)* | *Lavandula angustifolia* | YD, EOC, CC, AO, Flv, Phn | 8 | Greenhouse |
| 2 | *Aslani et al. (2023)* | *Salvia officinalis* | YD, EOC, EOY, CC, RWC | 3 | Greenhouse |
| 3 | *Celikcan, Kocak & Kulak (2021)* | *Ocimum basilicum* L. | YD | 3 | Greenhouse |
| 4 | *Rahimi et al. (2023a)* | *Thymus vulgaris* L. | YD, EOC, EOY, CC, EAO | 3 | Field |
| 5 | *Rahimi et al. (2023b)* | *Ocimum basilicum* | YD, EOC, CC, Flv, Phn, Pro. | 6 | Greenhouse |
| 6 | *Abdi et al. (2018)* | Mentha × piperita L. | YD, AO, Flv, Phn | 6 | Greenhouse |
| 7 | *Karimi et al. (2020)* | *Salvia officinalis* L. | YD, EOC, Pro | 4 | Greenhouse |
| 8 | *Govahi et al. (2015)* | *Salvia officinalis* L. | YD, EOC, EOY | 3 | Field |
| 9 | *Mulugeta, Sárosi & Radácsi (2023)* | *Ocimum basilicum* | YD, EOC, EOY, CC, RWC, AO, Phn | 3 | Greenhouse |
| 9 | *Mulugeta, Sárosi & Radácsi (2023)* | Ocimum africanum | YD, EOC, EOY, CC, RWC, AO, Phn | 3 | Greenhouse |
| 9 | *Mulugeta, Sárosi & Radácsi (2023)* | Ocimum americanum | YD, EOC, EOY, CC, RWC, AO, Phn | 3 | Greenhouse |
| 10 | *Bettaieb et al. (2011a)* | *Cuminum cyminum* L. | YD, CC, AO, Phn | 3 | Greenhouse |
| 11 | *Mulugeta & Radácsi (2022)* | *O. basilicum* 'Ohre' | YD, EOC, EOY, AO, Phn | 2 | Field |
| 11 | *Mulugeta & Radácsi (2022)* | *O. basilicum* 'Genovese' | YD, EOC, EOY, AO, Phn | 2 | Field |
| 11 | *Mulugeta & Radácsi (2022)* | O. × africanum | YD, EOC, EOY, AO, Phn | 2 | Field |
| 11 | *Mulugeta & Radácsi (2022)* | *O. americanum* | YD, EOC, EOY, AO, Phn | 2 | Field |
| 11 | *Mulugeta & Radácsi (2022)* | O. selloi | YD, EOC, EOY, AO, Phn | 2 | Field |
| 11 | *Mulugeta & Radácsi (2022)* | O. sanctum 'Krishna' | YD, EOC, EOY, AO, Phn | 2 | Field |
| 12 | *Bayat & Moghadam (2019)* | *Salvia nemorosa* L. Isfahan | YD, CC, FLV, Phn | 6 | Greenhouse |
| 13 | *Razmjoo, Heydarizadeh & Sabzalian (2008)* | Matricaria chamomila | YD | 3 | Field |
| 14 | *García-Caparrós et al. (2019)* | *L. latifolia* | YD, EOC, EOY, RWC | 4 | Field |
| 14 | *García-Caparrós et al. (2019)* | *M. piperita* | YD, EOC, EOY, RWC | 4 | Field |
| 14 | *García-Caparrós et al. (2019)* | S. lavandulifolia | YD, EOC, EOY, RWC | 4 | Field |
| 14 | *García-Caparrós et al. (2019)* | S. sclarea | YD, EOC, EOY, RWC | 4 | Field |
| 14 | *García-Caparrós et al. (2019)* | T. capitatus | YD, EOC, EOY, RWC | 4 | Field |
| 14 | *García-Caparrós et al. (2019)* | T. mastichina | YD, EOC, EOY, RWC | 4 | Field |
| 15 | *Hassanpour et al. (2014)* | *Mentha pulegium* | YD, EOC | 4 | Greenhouse |
| 16 | *Begum et al. (2021)* | *Nicotiana tabacum* | YD | 4 | Greenhouse |
| 17 | *Mohammadi, Ghorbanpour & Brestic (2018)* | *Thymus vulgaris* | YD, EOC, CC, RWC, EOC, Pro | 3 | Field |
| 18 | *Emami Bistgani et al. (2017)* | Thymus daenensis | YD, EOC, EOY, CC, Pro | 3 | Field |
| 19 | *Bettaieb et al. (2011b)* | *Salvia officinalis* L. | YD, CC, AO, Phn, | 3 | Greenhouse |
| 20 | *Habibi et al. (2024)* | Aloe Vera L. | YD, RWC, AO, Flv, Phn, Pro | 8 | Greenhouse |
| 21 | *Torun et al. (2021)* | *Hypericum perforatum* L. | RWC, EAO, Flv, Pro | 6 | Greenhouse |
| 22 | *Talbi et al. (2020)* | Oudeneya africana | Phn, FLV, AO, RWC | 6 | Greenhouse |
| 23 | *Bidabadi, VanderWeide & Sabbatini (2020)* | *Salvia nemorosa* L. | EOC, EOY, EAO | 3 | Greenhouse |
| 24 | *Amiri et al. (2017)* | *Pelargonium graveolens* L. | EOC, RWC, AO, FLV, Phn, Pro | 3 | Greenhouse |

| Ref. no | References | Plant species | Parameters | Repetition | Study condition |
|---|---|---|---|---|---|
| | | | **Table 2 (continued)** | | |
| 25 | *Imani et al. (2023)* | Salvia miryazanii | CC, EAO, Proline | 3 | Greenhouse |
| 26 | *Jafari & Shahsavar (2021)* | C. Aurantifolia swingle | FLV, Phn | 4 | Greenhouse |
| 27 | *Rahimi et al. (2022)* | *Thymus vulgaris* L. | EOC, EOY, CC, RWC, EAO, FLV, Phn, Pro. | 3 | Field |

**Note:**
YD, Yield; EOC, Essential oil content; EOY, Essential oil yield; CC, Chlorophyll Content; RWC, Relative water content; AO, Antioxidant; EAO, Enzymatic antioxidant; Flv, Flavonoid; Phn, Phenols; Pro, Proline.

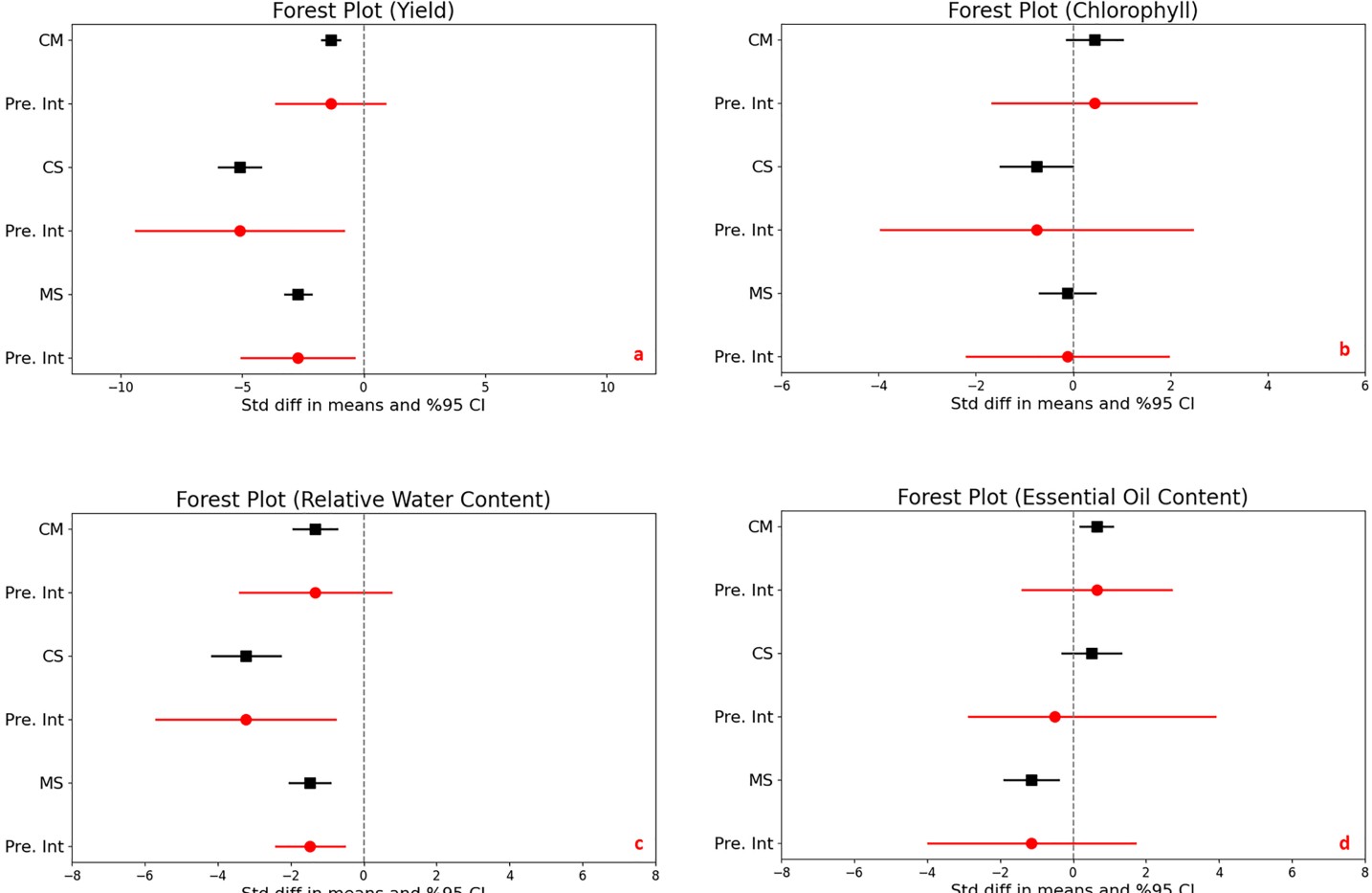

**Figure 2** Mean effect size changes in yield (A), chlorophyll (B), relative water content (C) and essential oil content (D) under drought stress compared to CM, Control *vs.* Moderate Stress; CS, Control *vs.* Severe Stress and MS, Moderate *vs.* Severe Stress. The horizontal bar indicates the 95% confidence interval (CI). The mean effect size line (CM, CS and MS) that does not overlap 0 indicates a significant increase at $p < 0.05$. "Pre. Int." indicate prediction intervals which show variation within the experiment population.

chlorophyll content across different studies. The prediction interval ranged from −1.662 to 2.533, demonstrating a broad spectrum of potential true effects and substantial variability in chlorophyll responses to stress across various populations.

## Relative water content

Significant reductions in relative water content were observed in plants subjected to moderate and severe drought stress compared to control conditions, clearly demonstrating the impact of environmental stress on plant hydration. Under moderate stress, there was a notable decrease with a mean effect size of −1.329 (95% CI [−1.927 to −0.730], Z = −4.352, $p < 0.001$). This reduction became even more pronounced under severe stress, with a mean effect size of −3.226 (95% CI [−4.165 to −2.286], Z = −6.727, $p < 0.001$). The comparison between moderate and severe stress levels revealed a significant further reduction in water content, emphasizing the compounded effect of increased stress severity (Fig. 2C).

The heterogeneity in the studies was moderate, with $I^2$ values ranging from 42% to 56%, and $p$-values between <0.001 to 0.002, indicating variability in how different studies measured the impact of drought on plant water content. This suggests that the response to drought might vary depending on specific plant species, environmental conditions, or experimental design.

Furthermore, the prediction intervals indicated a wide range of possible true effects, spanning from −3.402 to 0.744 for moderate stress and displaying even more substantial variability under severe stress. These intervals highlight the critical impact of drought on plant water content and underscore the significant variability across different environments and conditions, reflecting the complex dynamics of plant responses to varying levels of drought stress.

## Essential oil content

The statistical analysis exploring the impact of drought stress on essential oil content in plants revealed varying effects across stress levels. Under moderate stress compared to control conditions, there was a significant increase in essential oil content, with a mean effect size of 0.641 (95% CI [0.206 to 1.077]) and a Z-score of 2.885, achieving significance ($p = 0.004$). In contrast, no significant change was observed between control and severe stress conditions, with a mean effect size of 0.515 (95% CI [−0.290 to 1.320]) and a Z-score of 1.255 ($p = 0.210$). A significant reduction in essential oil content was noted when comparing moderate to severe stress, with a mean effect size of −1.137 (95% CI [−1.880 to −0.395]) and a Z-score of −3.002 ($p = 0.003$) (Fig. 2D).

High heterogeneity was evident across the studies ($I^2$ ranging from 71% to 82%, $p < 0.001$), suggesting diverse effects of drought on essential oil content. Additionally, the prediction intervals, spanning −1.402 to 2.684 for moderate stress and −2.863 to 3.894 for severe stress, highlight the substantial and unpredictable variability in how drought impacts essential oil production across different populations.

## Essential oil yield

For the control *vs*. moderate stress comparison, the mean effect size is −0.152 with a 95% confidence interval ranging from −0.982 to 0.677. The statistical test (Z-value = −0.360, $p = 0.719$) does not provide enough evidence to reject the null hypothesis that the mean effect size is zero, given an alpha of 0.050. For the control *vs*. severe stress comparison, the mean effect size is significantly negative at −2.899 with a 95% confidence interval of −4.454

to −1.344. The related Z-value is −3.654 with a *p*-value less than 0.001. This result allows us to reject the null hypothesis at the 0.050 alpha level, indicating a significant mean effect size that is not zero. For the moderate *vs.* severe stress comparison, the mean effect size is also significantly negative at −3.207 with a 95% confidence interval of −4.810 to −1.605. With a Z-value of −3.922 and a *p*-value less than 0.001, we again reject the null hypothesis at the 0.050 alpha level, concluding that the mean effect size is significantly different from zero (Fig. 3A).

High heterogeneity was evident across the studies ($I^2$ ranging from 76% to 80%, $p < 0.001$), suggesting diverse effects of drought on essential oil yield. The I-squared statistic is ranged from which tells us that some the variance in observed effects reflects variance in true effects rather than sampling error. Also, the prediction interval is ranged from −3,908 to 3,603 for CM, is −8,321 to 2,522 for CS and is −9,001 to 2,587 for MS which how much variance across to population.

### Non enzymatic antioxidants (DPPH, FRAP, ABTS)

For the control *vs.* moderate stress comparison, the mean effect size is −0,091 with a 95% confidence interval of −0,696 to 0,514. The mean effect size in the universe of comparable studies could fall anywhere in this interval. The Z-value tests the null hypothesis that the mean effect size is zero. The Z-value is −0,295 with $p = 0,768$. Using a criterion alpha of 0,050, we cannot reject this null hypothesis. For the control *vs.* severe stress comparison, the mean effect size is −0,226 with a 95% confidence interval of −1,183 to 0,732. The mean effect size in the universe of comparable studies could fall anywhere in this interval. The Z-value tests the null hypothesis that the mean effect size is zero. The Z-value is −0,462 with $p = 0,644$. Using a criterion alpha of 0,050, we cannot reject this null hypothesis. For the moderate *vs.* severe stress comparison the mean effect size is −0,451 with a 95% confidence interval of −1,157 to 0,255. The mean effect size in the universe of comparable studies could fall anywhere in this interval. The Z-value tests the null hypothesis that the mean effect size is zero. The Z-value is −1,252 with $p = 0,211$. Using a criterion alpha of 0,050, we cannot reject this null hypothesis (Fig. 3B).

High heterogeneity was evident across the studies ($I^2$ ranging from 73% to 85%, $p < 0.001$), suggesting diverse effects of drought on non-enzymatic antioxidants. The I-squared statistic is ranged from which tells us that some the variance in observed effects reflects variance in true effects rather than sampling error. Also, the prediction interval is ranged from −2.779 to 2.597 for CM, is −4.185 to 3.734 for CS and is −3.314 to 2.412 for MS which how much variance across to population.

### Enzymatic antioxidants

The analysis investigated the effects of stress on enzymatic antioxidant levels in plants, producing significant results across different stress conditions. There was a significant increase in enzymatic antioxidants when comparing control to moderate stress, with a mean effect size of 4.017. The 95% confidence interval (CI) ranged from 1.184 to 6.849, and statistical significance was confirmed by a Z-score of 2.779 and a *p*-value of 0.005, indicating a robust response to moderate stress. Control *vs.* severe stress, a increase were

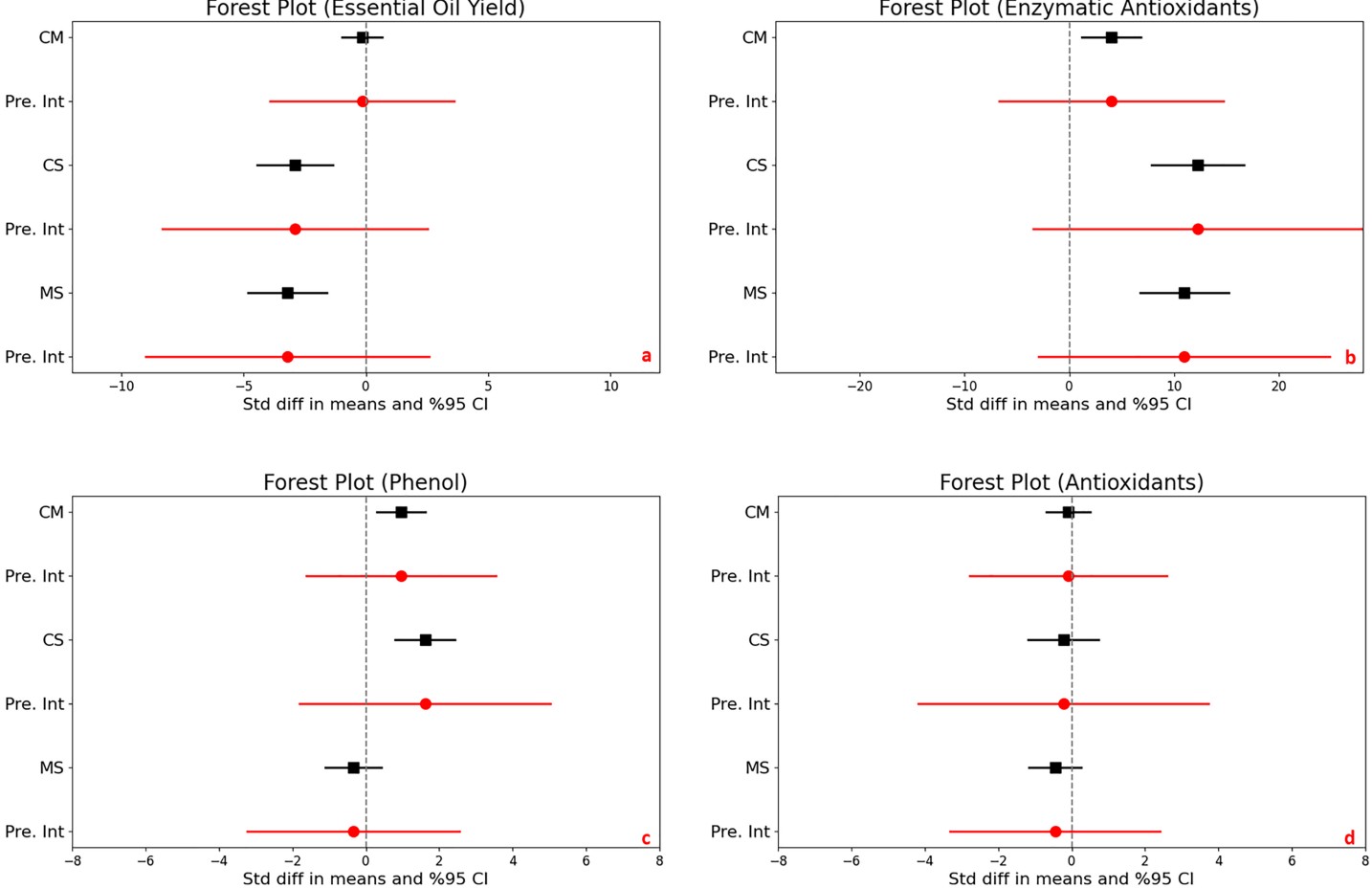

**Figure 3  Mean effect size changes in essential oil yield (A), enzymatic antioxidants (B), phenol (C) and antioxidants (D) under drought stress compared to CM, Control *vs.* Moderate Stress; CS, Control *vs.* Severe Stress and MS, Moderate *vs.* Severe Stress.** The horizontal bar indicates the 95% confidence interval (CI). The mean effect size line (CM, CS and MS) that does not overlap 0 indicates a significant increase at $p < 0.05$. "Pre. Int." indicate prediction intervals which show variation within the experiment population. 

determined under severe stress conditions, with a mean effect size of 12.248 and a 95% CI from 7.834 to 16.662. The Z-score of 5.439 and a *p*-value less than 0.001 suggest a substantial increase in antioxidant levels due to severe stress. Comparing moderate to severe stress, a significant difference was noted, with a mean effect size of 10.962 and a 95% CI ranging from 6.730 to 15.193. The Z-score was 5.078 with a *p*-value less than 0.001, demonstrating a significant escalation in enzymatic antioxidant production from moderate to severe stress levels (Fig. 3C).

Heterogeneity was extremely high under both controls to moderate and control to severe stress conditions, with $I^2$ values of 90% and 87%, respectively, and corresponding Q-values of 136.786 and 91.476 (both $p < 0.001$). This indicates significant variability in the antioxidant responses to drought across studies.

Prediction intervals show a wide range of potential true effects, from −6.693 to 14.726 under moderate stress and from −3.418 to 27.914 under severe stress, highlighting the

broad variability and complex dynamics of antioxidant response to different levels of stress. This wide range suggests that while antioxidants generally increase with stress, individual responses can vary dramatically.

### Phenols and flavonoids

In assessing the impact of drought stress on the production of phenols and flavonoids in plants, significant increases were observed under varying stress conditions. Specifically, under moderate stress, phenols increased significantly with a mean effect size of 0.958 (95% CI [0.299 to 1.616], Z = 2.850, $p$ = 0.004), and flavonoids showed an even larger increase with a mean effect size of 2.277 (95% CI [0.932 to 3.623], Z = 3.317, $p$ = 0.001). The comparison between moderate and severe stress conditions further highlighted that severe stress significantly increases the accumulation of both compounds (Figs. 3D and 4A).

High levels of heterogeneity in the responses of phenols ($I^2$ = 76%) and flavonoids ($I^2$ = 88%) to drought stress ($p < 0.001$) suggest substantial variability across different studies. The prediction intervals also reflected significant variability, ranging from −1.618 to 3.533 for phenols and −1.903 to 6.458 for flavonoids under moderate stress, illustrating the broad spectrum of possible true effects of drought on these antioxidant compounds across diverse populations.

### Proline content

Significant increases in proline content were noted under both moderate and severe stress conditions compared to control, highlighting a substantial trigger proline accumulation as a stress response. The mean effect sizes provided clear evidence of this increase, although specific values were not detailed. The analysis also uncovered high heterogeneity across the studies, with $I^2$ values ranging from 73% to 79% and $p$-values below 0.001, indicating that the extent of proline accumulation varies significantly among different studies (Fig. 4B). This suggests diverse physiological responses to drought stress depending on environmental factors or experimental conditions. Furthermore, the wide prediction intervals, ranging from −1.505 to 5.139 for moderate stress and from −1.406 to 4.111 for severe stress, underscore the substantial variability in proline responses across different environments. These intervals reflect a broad spectrum of possible effects, from decreases to substantial increases in proline content under varying degrees of drought stress.

### DISCUSSION

This meta-analysis uses 27 studies to explores the effects of drought on agronomic, physiological, and biochemical properties, with a specific focus on medicinal plants. In the studies of plant responses to abiotic stress, several parameters are commonly used to assess the tolerant and adaptability of plants. Among these, proline and enzymatic antioxidants are frequently examined due to their critical roles in plant defense mechanisms. For clarity and systematic discussion, each parameter is analyzed in separate subsections, providing a structured presentation of the findings.

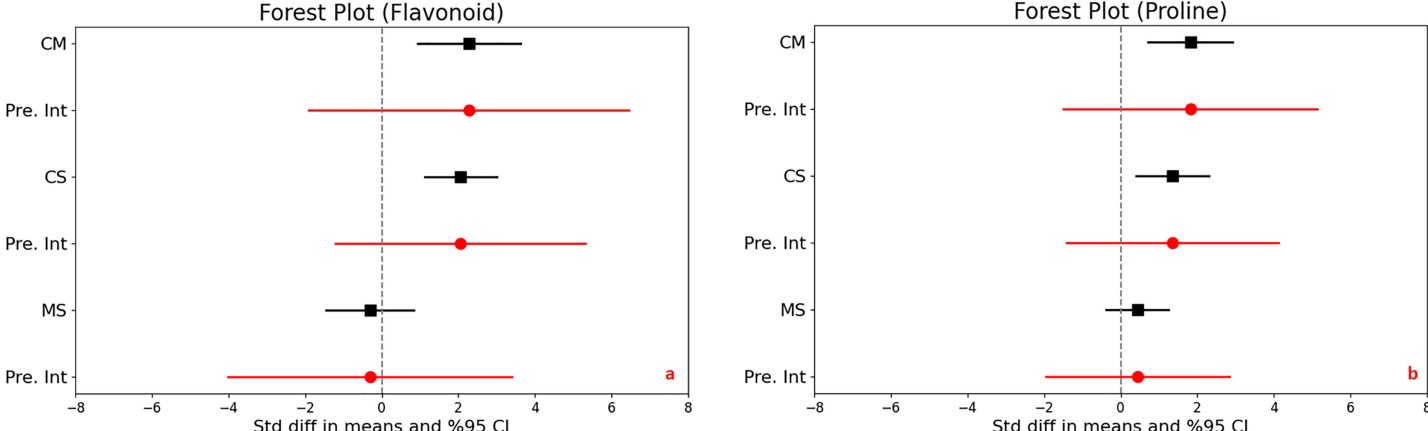

**Figure 4 Mean effect size changes in flavonoid (A), proline (B) under drought stress compared to CM, Control *vs*. Moderate Stress; CS, Control *vs*. Severe Stress; and MS, Moderate *vs*. Severe Stress.** The horizontal bar indicates the 95% confidence interval (CI). The mean effect size line (CM, CS, and MS) that does not overlap 0 indicates a significant increase at $p < 0.05$. "Pre. Int." indicates prediction intervals that show variation within the experiment population.

## Plant yield or plant biomass

According to meta-analysis results, drought stress has been statistically proven to significantly reduce plant yield in comparisons between control (no stress) *vs*. moderate stress, control *vs*. severe stress, and moderate *vs*. severe stress. Effect size is most affected during transitions involving severe stress which can be said that below 50% field capacity has significant negative effect on plant yield.

It is common for plant yield to decrease under drought conditions. Many studies have reported reductions in plant yield or biomass due to drought stress (*Chrysargyris et al., 2016*; *Aslani et al., 2023*; *Celikcan, Kocak & Kulak, 2021*; *Rahimi et al., 2023a*, *2023b*; *Abdi et al., 2018*; *Karimi et al., 2020*). There are multiple mechanisms for drought stress to significantly decreases plant yield and biomass. Water deficit inhibits $CO_2$ assimilation by causing stomatal closure, membrane deterioration, and enzyme dysfunction, which in turn reduces the production of carbohydrates necessary for plant growth (*Farooq et al., 2011*). Physiologically, drought leads to smaller leaves, shorter stems, and stunted root development, as plants prioritize survival over growth, further decreases biomass production (*Begna, 2020*).

## Chlorophyll content

When examining chlorophyll values, it has been found that only under control *vs*. severe stress conditions, there is a statistically significant decrease in chlorophyll at $p < 0.05$. Meta-analysis indicates that chlorophyll is not significantly affected under control *vs*. moderate and moderate *vs*. severe stress conditions. Generally, it is expected that chlorophyll levels decrease with stress, as many studies report reductions in chlorophyll under drought conditions (*Aslani et al., 2023*; *Rahimi et al., 2023a*, *2023b*, *Imani et al., 2023*). However, there are also few other studies suggests opposite that chlorophyll
increases under drought stress (*Mulugeta, Sárosi & Radácsi, 2023*; *Emami Bistgani et al., 2017*).

The key mechanisms related to chlorophyll changes under drought stress is, stress leads to a reduction in chlorophyll synthesis, essential for photosynthesis, by affecting its biosynthetic pathways. Stomatal closure is a common plant response to minimize water loss during drought conditions. This adaptive mechanism, while conserving water, also reduces the internal $CO_2$ concentration, negatively impacting photosynthetic efficiency. Consequently, the lower photosynthetic activity results in decreased energy and substrates necessary for chlorophyll production, further diminishing chlorophyll a and b levels, crucial for capturing light energy (*Meher, Singh & Chawla, 2018*; *Farooq et al., 2011*). The decreases of biosynthesis process are insufficient to explain the reduction in chlorophyll levels. Other factors must also be considered to fully understand the decline in chlorophyll. Drought stress leads to an increase in reactive oxygen species (ROS) within plant cells, causing oxidative damage to chloroplasts and chlorophyll molecules. This damage not only disrupts chlorophyll-protein complexes that are vital for light absorption but also impairs the structural integrity of chloroplasts, including their membrane systems. Such disruptions reduce both the synthesis of new chlorophyll and the stability of existing chlorophyll, ultimately leading to a decrease in chlorophyll content in drought-stressed plants. This affects the plant's ability to perform photosynthesis efficiently (*Alberte & Thornber, 1977*) and *Zhang et al. (2019)*. However, there are studies that suggest the contrary as we mentioned before.

For instance, *Portulaca oleracea* exhibits an increase in chlorophyll under drought, suggesting an adaptive mechanism that enhances drought resilience (*Rahdari, Hosseini & Tavakoli, 2012*). On the other hand, *Arachis hypogaea* show a decrease in chlorophyll with drought stress, but certain genotypes maintain chlorophyll stability, indicating that genetic factors play role by preserving chlorophyll content (*Arunyanark et al., 2008*).

This meta-analysis indicates that there is no significant change in chlorophyll levels under drought stress, except for a decrease noted during the transition from control to severe stress, with a $p$-value of 0.047. However, given the heterogeneity of the study, this result's reliability is uncertain. There could be various reasons why chlorophyll levels did not show significant changes according to our meta-analysis results.

*1) Leaf position* (The location of the leaf where measurements are taken): The variation in chlorophyll content loss between upper and lower leaves during drought stress is linked to physiological and morphological differences. Lower leaves, which are older, experience a carbon/nitrogen imbalance with higher carbon to nitrogen ratios under drought conditions, leading to earlier and more pronounced senescence and chlorophyll degradation (*Chen et al., 2015*). This senescence plays a critical role in nutrient remobilization, helping to redistribute nutrients to younger tissues, thus aiding plant survival under stress. Research highlights that these changes are particularly notable in lower leaves, which are more susceptible to this carbon/nitrogen imbalance, thus demonstrating a differential response that supports the plant's adaptation and survival in limited water conditions (*Chen et al., 2015*). So, if measurements are taken from the upper parts of leaves, it is more likely that no changes in chlorophyll values are observed.

*2) Stress duration* (the length of time the plants are subjected to stress): The duration of abiotic stress such as drought stress can vary from hours to months or longer under the natural field conditions. The plant responses to abiotic stress likely vary depending on the duration of stress. Consistent with this notion, a recent study showed that distinct gene networks drive differential response to abrupt or gradual long-term water deficit (*Ambrosone et al., 2017*). According to *Heidari, Bandehagh & Toorchı (2014)* study short-term stress initially increased chlorophyll content due to secondary effects, but prolonged exposure led to a decline. This indicates that plants may temporarily boost chlorophyll to enhance photosynthesis under moderate stress, but sustained high levels of stress eventually deteriorate the photosynthetic apparatus. Extended periods of water deficiency can lead to significant structural damage to the photosystem proteins and the thylakoid membranes where chlorophyll is housed. This structural damage is often irreversible, leading to a sustained decrease in photosynthetic efficiency even after rewatering (*Li et al., 2020*).

In conclusion, according to the results of meta-analyses, chlorophyll content values are not significantly affected by drought. Moreover, this may be due to various factors such as genetic factors, the position of the leaves from which samples were taken, and the duration of stress. These factors require further detailed investigation.

## Relative water content

According to the meta-analysis results, the relative water content showed a statistically significant negative change in effect size across all stress conditions, with a *p*-value less than 0.01.

Studies have shown that under drought conditions, the RWC of plants tends to decrease significantly, reflecting the impact of water stress on plant hydration levels (*Wang et al., 2019*; *Mukami et al., 2019*; *Valentovič et al., 2006*; *Saraswat et al., 2020*). Additionally, the decrease in RWC is associated with an increase in osmotic potential within plant cells, indicating the cellular response to water deficit (*Wang et al., 2019*). Research has highlighted the importance of RWC as a reliable indicator for assessing both the sensitivity and tolerance of plants to water deficit conditions (*Saraswat et al., 2020*). Maintaining higher RWC levels has been linked to better drought tolerance, as evidenced by studies showing that genotypes with higher leaf water potential and RWC exhibit higher photosynthetic rates under drought stress (*Sağlam, Terzi & Demiralay, 2014*).

## Essential oil content

When examining essential oil ratios, it was found that under control *vs.* moderate stress, the essential oil ratios increased. However, under control *vs.* severe stress, the meta-analysis results indicated that the essential oils did not exhibit a significant effect size. Under moderate *vs.* severe stress, the effect size value was statistically significant with a $p < 0.01$, showing a negative impact.

When examining the literature, there are two differing opinions regarding essential oil content. One view argues that the essential oil ratios in medicinal plants increase under abiotic stress (*Govahi et al., 2015*; *Mohammadi, Ghorbanpour & Brestic, 2018*). Conversely,

another view claims that essential oil ratios decrease (*Aslani et al., 2023*, *Rahimi et al., 2023a*; *Imani et al., 2023*). Currently, the more widely accepted view is that essential oil content increases under stress conditions. However, meta-analysis results suggest that both views can be true depending on specific conditions. It has been determined that essential oil ratios increase under moderate levels of stress. Yet, as stress levels intensify, a decrease in essential oil content is observed. This pattern is evident that essential oil ratios increase under control *vs.* moderate stress but do not change significantly under control *vs.* severe stress. This finding becomes more apparent when comparing moderate *vs.* severe stress, showing a significant reduction in essential oil content.

The changes in essential oil content under abiotic stress conditions are attributed to modifications in biosynthetic processes (*Ghassemi-Golezani & Rahimzadeh, 2022*). For example, stress has been demonstrated to impact the content of essential oil components by altering biosynthetic pathways (*Ghassemi-Golezani & Rahimzadeh, 2022*). Additionally, the increase in essential oil content under stress conditions is associated with the stimulation of biosynthesis, highlighting a complex relationship between stress responses and secondary metabolite production (*Neto et al., 2019*). Another reason is, the response of essential oil content to drought stress varies among plant species or genotypes of same species. For instance, severe drought stress has been found to negatively affect the essential oil content, essential oil yield, and antioxidant capacity in *Ocimum* species (*Musie & Radácsi, 2022*). Conversely, in *Thymus vulgaris* L., essential oil content increases under slight drought stress conditions, but severe drought stress can reduce the percentage of essential oil due to factors such as essential oil storage and reduction in leaf area (*Rahimi et al., 2023b*).

The results of these studies are consistent with the meta-analysis findings. Both researchers' results and the meta-analysis support the observation that essential oil content increases under moderate stress conditions but decreases during severe drought stress.

## Essential oil yield

According to the meta-analysis results, while essential oil yield does not show statistical significance under control *vs.* moderate stress, the effect size value significantly decreases under control *vs.* severe and moderate *vs.* severe stress, with $p$-values less than 0.01. This reduction in essential oil yield aligns with the changes observed in essential oil content due to drought stress.

Essential oil yield is a parameter influenced by both yield and essential oil content. According to the meta-analysis results, under control *vs.* moderate stress, an increase in essential oil content coupled with a decrease in yield has resulted in a neutral effect on essential oil yield. However, in the other two stress comparisons control *vs.* severe and moderate *vs.* severe show a decrease in essential oil content and yield have contributed to a reduction in essential oil yield. Other studies have also shown that drought stress can reduce essential oil yield in medicinal plants. For instance, *Saeidnejad et al. (2013)* found a decrease in essential oil yield in peppermint (*Mentha piperita* L.) as drought stress become more severe. Similarly, *Gholinezhad (2020)* observed that increasing drought stress led to a reduction in essential oil yield in pot marigold (*Calendula officinalis* L.). Additionally,

*Musie & Radácsi (2022)* reported a significant reduction in essential oil yield of *Ocimum* species under severe drought conditions, with a 50% loss in both fresh and dry herb yield.

## Non enzymatic antioxidants (DPPH, FRAP, ABTS)

According to *Chrysargyris et al. (2016)* and *Talbi et al. (2020)* studies on medicinal plants suggest that non-enzymatic antioxidants serve as a defense mechanism against reactive oxygen species in response to drought stress and shows increase as part of the plant's adaptive strategy to mitigate stress-related damage. Also, different results can be seen in *Abdi et al. (2018)* and *Bettaieb et al. (2011a)*, in which non-enzymatic antioxidants decrease with drought stress.

However, our meta-analysis indicates that across all levels of stress (control, moderate, and severe), the mean effect size was not statistically significant. This suggests that it is not accurate to claim that non-enzymatic antioxidants increase under drought stress. Generally, the results can vary from one plant to another, but based on our meta-analysis, it appears that drought stress does not significantly alter these antioxidants. Therefore, relying on these antioxidants as indicators of drought resistance would be misleading according to the results.

Thyme plants subjected to different irrigation intervals showed that severe water stress significantly reduced antioxidant activity as measured by DPPH and FRAP assays. Long-term drought led to decreased total phenolic content, indicating overall lower antioxidant activity under water stress conditions (*Khosh-khui, Ashiri & Saharkhiz, 2012*). Research on various plant species under drought conditions has shown that different species and different conditions of drought can lead to varying responses in antioxidant activities. Some plants may exhibit increased antioxidant activities as a defense mechanism, while others might show a decline due to the severity of the stress (*Štajner et al., 2011*).

## Enzymatic antioxidants

Our meta-analysis results show that enzymatic antioxidants demonstrated a statistically significant mean effect size ($p < 0.01$) across all stress conditions, with a positive impact observed. Compared to non-enzymatic antioxidants, enzymatic antioxidants are more responsive and provide a more effective response against stress, as evidenced by the meta-analysis results.

Antioxidants are essential in plants' response to drought stress, helping to alleviate the harmful effects of ROS generated during oxidative stress. Studies have demonstrated that there is a dynamic change in the activities of antioxidative enzymes under drought conditions, which is a crucial mechanism for coping with oxidative stress (*Singh, Gupta & Kaur, 2011*). The upregulation of antioxidant enzymes serves as a significant marker for drought stress and contributes to drought tolerance (*Laxa et al., 2019*). Maintaining high antioxidant capacity and enzymatic activity during drought stress aids plants in acclimatizing and surviving stress conditions by neutralizing ROS, thereby enhancing drought tolerance (*Rao et al., 2020*).

## Phenols

According to our meta-analysis results, the phenolic content exhibited a statistically significant positive change in mean effect size under both control *vs.* moderate and control *vs.* severe stress conditions.

Total phenolic content in medicinal plants can exhibit varying responses to drought stress. While some studies have reported a decrease in total phenolic compounds under drought conditions in plant species like grapevines (*Król, Amarowicz & Weidner, 2014*), other research has shown an increase in total phenolic compounds in stressed plants, which can act as signal molecules and antioxidants (*AlKahtani et al., 2021*). Drought stress typically leads to an increase in "total phenolic content" in plants as a protective response against oxidative stress. This increase in phenolics is part of the plant's adaptive mechanism to cope with the reduced water availability (*Selmar & Kleinwächter, 2013*). The production and accumulation of phenolic compounds are part of a broader range of biochemical adaptations to drought, including increased osmolytes and antioxidant activities, which help plants manage the stress (*Abdelaal et al., 2021*). A study on canola (*Brassica napus*) showed that while drought stress led to various changes in biochemical markers, the response in phenolic content was dependent on the severity and the cultivar (*Rezayian, Niknam & Ebrahimzadeh, 2018*).

## Flavonoid

According to the meta-analysis results, the flavonoid levels showed a statistically significant positive change in effect size under both control *vs.* moderate stress and control *vs.* severe stress, with *p*-values less than 0.01.

Flavonoid levels in plants can exhibit dynamic changes in response to drought stress. Studies have shown that under drought conditions, there is an enhancement in flavonoid accumulation, which can contribute to plant resistance against stress factors (*Nakabayashi et al., 2013*; *Gargallo-Garriga et al., 2018*). Flavonoids with radical scavenging properties have been reported to mitigate oxidative stress and enhance drought tolerance in plants like *Arabidopsis thaliana* (*Nakabayashi et al., 2013*). Additionally, the presence of flavonoids such as acacetin and homoorientin has been noted to increase during drought stress, indicating a response to water stress (*Gargallo-Garriga et al., 2018*). Conversely, some studies have reported a decrease in flavonoid content under drought stress. For instance, low soil moisture levels have been associated with a significant reduction in total flavonoid content (*La et al., 2023*). Furthermore, drought stress has been linked to a decrease in flavonoid biosynthesis in *Polygonatum kingianum* tubers (*Qian et al., 2021*).

## Proline content

Proline serves as an osmolyte, antioxidant, and signaling molecule, aiding in stress tolerance and plant survival under water stress conditions (*Sharma & Agarwal, 2021*). The increase in proline content is a common adaptive response of plants to drought stress, as it helps in osmotic adjustment and defense against oxidative damage (*Sharma & Agarwal, 2021*). Studies have shown that proline accumulation is positively correlated with drought tolerance in plants (*Nasrin et al., 2020*). The accumulation of proline under
drought stress conditions is associated with enhanced osmoregulation, compatible solutes, and biochemical changes that contribute to stress tolerance (*Arivalagan & Somasundaram, 2016*). Additionally, the upregulation of genes involved in proline metabolism under drought stress further supports the role of proline in plant adaptation to water stress conditions (*Schafleitner et al., 2006*).

According to the meta-analysis results, when comparing control *vs.* moderate and control *vs.* severe stress, the effect size was found statistically increase in a positive direction ($p < 0.01$). However, no significant change was detected under moderate *vs.* severe stress. This suggests that proline predominantly becomes active under moderate stress levels and does not significantly change as stress levels increase. The synthesis of proline in response to drought stress often varies with the severity of the stress. Moderate drought stress generally stimulates proline accumulation significantly as part of an adaptive response to help maintain cellular osmolarity, protect cellular structures, and scavenge free radicals. However, under severe drought stress, this response may plateau or even decline due to severe damage to the plant's physiological and metabolic capacities (*Habibi et al., 2024*).

As summary:

The meta-analysis results revealed the effect sizes change across different comparisons: control compared to moderate stress, control compared to severe stress, and moderate compared to severe stress.

Parameters which have very high effect size (4 and above): Enzymatic antioxidants (peroxidase (POD), catalase (CAT), superoxide dismutase (SOD)) have very high effects size in three stress comparison (CM, CS and MS).

Parameters which have high effect size (2 to 4): Flavonoids have higher effect sizes under control compared to moderate stress. When comparing severe stress to control conditions, the highly impacted parameters include yield (YD), relative water content (RWC), essential oil yield (EOY), and flavonoids. The transition from moderate to severe stress shows the most significant effects on essential oil components EOC, EOY, and yield (YD).

Parameters which have medium effect size (1 to 2) change: Under moderate stress compared to control, the effected parameters were proline, yield and RWC. Under severe stress compared to control, phenols and proline show moderate changes. RWC and EOC shows moderate changes when comparing moderate to severe stress conditions.

Parameters with low effect size (0–1) under different stress comparisons: Under moderate stress compared to control phenols show a positive change, EOC shows a positive change, while chlorophyll shows no significant change. Under severe stress compared to control, chlorophyll, EOC and antioxidant (AO) shows no significant changes with low effect sizes. From moderate compared to severe stress: AO, proline, phenols, flavonoids, and chlorophyll all show no significant changes with low effects sizes.

According to the results of the meta-analysis, enzymatic antioxidants play a highly active role in all three stress conditions, demonstrating a significant response to drought. Specifically, the activities of enzymatic antioxidants such as superoxide dismutase (SOD), peroxidase (POD), and catalase (CAT) are observed to change responsively under drought stress. In contrast, non-enzymatic antioxidants (DPPH, FRAP, ABTS) were found to be

insufficient in all three stress conditions, showing no statistically significant changes. Although non-enzymatic antioxidants are often proposed as part of plants' adaptation mechanisms to abiotic stress, the meta-analysis suggests that this may not hold true for drought conditions.

Contrary to claims in various studies, chlorophyll content has been identified as another parameter that remains unaffected by drought stress. Although the effect size was initially significant at −0.752 with a $p$-value of 0.047 in the comparison between control and severe stress conditions, this significance diminishes when considering heterogeneity and after recalculating using the "trim and fill method" for an unbiased estimate. Therefore, the meta-analysis results indicate that chlorophyll levels are not statistically affected in any of the three stress conditions. While it may seem logical from both physiological and theoretical perspectives to expect a decrease in chlorophyll content under stress, the reasons why this is not the case are inspected and discussed in the discussion section.

Another significant aspect is the content of essential oils. The meta-analysis reveals that essential oil content shows a statistically significant but small positive mean effect size change under moderate stress compared to control stress conditions. However, as the level of stress increases, there is a shift towards a negative change. Specifically, when transition from moderate to severe stress, there is a medium-sized negative effect observed. This suggests that essential oil levels may initially increase under moderate stress conditions but begin to decrease as the stress level intensifies.

Proline, flavonoid, and phenol contents have been identified as statistically increasing in a positive direction when comparing control to both moderate and severe stress conditions. However, no significant change in these values were observed when comparing moderate to severe stress. Another critical parameter, relative water content, has shown a significant negative change in all three stress comparisons. Additionally, yield and essential oil yield which productivity-related parameters, have been found to trend downward as stress levels increase.

# CONCLUSION

According to meta-analysis results, enzymatic antioxidants (SOD, POD, CAT) emerge as the most responsive parameters to stress. It is well-known that these antioxidants enhance plant tolerance against abiotic stress factors. In this study, when compared to other parameters based on effect size values, enzymatic antioxidants can be considered the most actively used defense mechanism by plants in response to drought stress. Therefore, these parameters are essential for assessing the effects of stress factors on plants and should be considered a priority in such evaluations.

In addition to the significant role of enzymatic antioxidants, other parameters such as relative water content (RWC) and yield also exhibit considerable negative effect sizes under all stress conditions. Therefore, when evaluating the impacts of drought stress on plants, it is beneficial to include these three attributes (enzymatic antioxidants, RWC, and yield) in to the studies. These parameters consistently show statistical significance and high effect sizes across all stress levels, indicating their utility in determining plant tolerance. Measuring these parameters can provide better insights than others, especially when

identifying tolerance genotypes, plant species, and/or developing tolerance to stress conditions through various interventions.

The meta-analysis results indicate that chlorophyll is not a reliable indicator for measuring stress tolerance mechanisms. It is worth noting that the duration of stress could be a significant factor, but due to the lack of long-term stress studies, the negative impact on chlorophyll may not have been observed in our meta-analysis. Parameters which important for medicinal plants, such as essential oil content, flavonoids and phenols have shown positive changes under control *vs*. moderate (CM) and control *vs*. severe (CS) stress, whereas antioxidant capacity (AO) remained unchanged. This suggests that antioxidant parameters like phenols and flavonoids are more affected compared to non-enzymatic antioxidant assays (DPPH, FRAP, and ABTS). Therefore, in plant stress studies, measuring antioxidants such as flavonoids and phenols could be a better option than using radical scavenging methods like DPPH, FRAP, and ABTS.

### Funding
The authors received no funding for this work.

### Competing Interests
The authors declare that they have no competing interests.

### Author Contributions
- Uğur Tan conceived and designed the experiments, performed the experiments, analyzed the data, prepared figures and/or tables, and approved the final draft.
- Hatice Kübra Gören analyzed the data, prepared figures and/or tables, authored or reviewed drafts of the article, and approved the final draft.

### Data Availability
This is a systematic review/meta-analysis.

### Supplemental Information
Supplemental information for this article can be found online at http://dx.doi.org/10.7717/peerj.17801#supplemental-information.

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
