# Peer review of "Comprehensive evaluation of drought stress on medicinal plants: a meta-analysis"

_PeerJ, doi:10.7717/peerj.17801_

## Round 0.1 · original submission · Major Revisions

The reviewers have noted areas of your manuscript that need correction. I would like you to revise your manuscript more carefully and in accordance with the journal format, especially due to the large number of spelling errors. Additionally, more information should be given about medicinal plants. In this form, the content of the article is incompatible with the title. Since medicinal plants are what makes your study unique, you should discuss the effects of drought stress by focusing on medicinal plants rather than general information. Customize the introduction section as well.

Reviewer 1 ·

Basic reporting

"Abstract"
Please provide additional explanations for the reasons why medicinal plants were examined in this study.

"Introduction"
Many studies have reported on the relationship between drought stress and plants, and summarizing these studies is very important for understanding the impact of climate change on crop production and the global environment. This study analyzes the relationship between drought stress and medicinal plants. However, most of the existing descriptions are only about crops. If there are similar studies on crops and wild plants that have already been reported, please cite them and provide a thorough explanation of why this study focuses on medicinal plants. For example, you could mention that a certain component contained in medicinal plants increases under drought stress.

Experimental design

"Bibliographic research and data collection"
In this paper, the degree of stress is evaluated based on field or soil water capacity, but these are merely cultivation conditions and do not indicate the actual degree of drought stress experienced by the plants. Shouldn't the drought stress state be evaluated using various water relations parameters (such as water potential at predawn or during the day)? Even under the same field or soil water capacity, the degree of water stress experienced by individual plants varies depending on the plant species or even the variety within the same species. This is a crucial point related to the analysis in this study. A reasonable explanation regarding this matter is necessary. In connection with the above, high heterogeneity in many parameters is shown in the Results section. Could this be related to using water capacity as the stress condition?
This paper mentions that parameters such as relative water content and proline are not frequently measured. Is this limited to medicinal plants? There are many previous studies on the water relations in plants. At what point is the relative water content measured? Is it the value at the turgor loss point? Rather, the water potential or osmotic potential at the turgor loss point might be more effective for evaluating the plant's response to drought stress.

Validity of the findings

As pointed out in "Bibliographic research and data collection," after providing a rational explanation for the evaluation of the degree of stress based on field or soil water capacity, we will assess the validity of the findings.

Additional comments

"Results"
It would be easier to understand if you add a table summarizing parameters such as I^2 and Q value, which indicate the heterogeneity among the studies. Although these values are mentioned in the text of the results section.

Line 84 Meta-analysis, essentially an ’analysis of analyses,’
Please include the references as follows. ’analysis of analyses,’(Glass,G.V. 1976 Primary, secondary and meta-analysis of research. Educational Researcher 5: 3-8)

Line 405 Rahimi et. al. 2023, Rahimi et. al. 2023,
The same reference is listed twice, so please delete one of them.

Figures 2 - 11
What is ‘pre.int’ an abbreviation for?

Table 1
What are ‘LL, UL and n’ abbreviations for?
(I know that ‘LL, UL and n’ are an abbreviation for lower limit, upper limit and number of samples, respectively.)

"References"
Some journal titles are in italics, while others are not. Please standardize them.

·

Basic reporting

The authors of this research article have undertaken a comprehensive study on the “Comprehensive evaluation of drought stress on medicinal plants: a meta-analysis” and tried to cover a detail and thorough analysis of how drought conditions affect medicinal plants. Tan and Hatice examined various aspects such as physiological and biochemical responses of these plants to water scarcity. By focusing on medicinal plants, the research emphasizes the implications of drought stress on plants that have significant therapeutic value. The comprehensive nature of the evaluation suggests that the study synthesizes existing research, potentially identifying patterns, critical insights, and knowledge gaps related to how drought impacts these valuable plant species. The methodology is clearly described. The results are quite interesting; however, the manuscript still needs some revision before accepted for final publication in this journal. Following points need to be addressed carefully;
1. There are a number of criticisms/weaknesses that need to be addressed to validate the arguments of the authors and to convey them effectively. In addition, the authors have several typos and grammatical errors in the manuscript and should use more careful writing. Therefore, it is recommended to thoroughly check the manuscript from top to bottom and make the corrections accordingly.
2. In abstract:
• Line-8: stated that limit their growth and productivity should be changed to growth and development.
• Line-18: Other parameters ……… ‘O’ should be in
• Authors need to use SI units for all the parameters and also better to mention the obtained results in number along with their units in all text but especially in the abstract. Also need explanation of the abbreviations first time used in abstract such as DPPH, FRAP and ABTS for general readers.
3. Keywords: Keywords should be in alphabetical order (A…….Z)
4. Manuscript should be uniformly arranged in terms of “units, titles, heading, formula, space in between the lines and font size etc.” as per the format of the journal. Why citations style is different from each other e.g. Line-43: poorly understood. (Kemp et al., 2022) and Line-49: (Asseng et. al. 2011)? Also there is no need of full stop before the citation as already mention above. In some citations authors wrote A & B while in some as A and B why? Authors are suggested to check the whole article thoroughly and make the corrections according the journal recommended format as given in the main webpage of the journal via name instruction to authors.
5. Although statistical analysis is appropriate, however, the authors need to mention about the replicates/repeats per treatment.
6. Why how did authors use these concentrations i.e. (100%-80%), (80%-50%), (below 50%).
7. Authors are suggested to use same format for the whole subheadings in the Results portion. As currently different alphabets styles are sued e.g. Line-249 is different from other subheadings why?
8. Add few more references in the discussion section relating to the response of medicinal plants to drought stress. Also check the citations style as currently are not in same format.
9. Line-454: 2) Stress condition………… what does it mean?.
10. All References should be in same and journal recommended format.
11. Figures quality is not good and need improvement because in current version all the figures are very blurred and not in readable form.

Experimental design

No comments

Validity of the findings

No comments

Additional comments

No Comments

Reviewer 3 ·

Basic reporting

Clear but needs to be specific on methods. See attached word document

Experimental design

Well defined

Validity of the findings

Relevant

Additional comments

See attached for more comments

Annotated reviews are not available for download in order to protect the identity of reviewers who chose to remain anonymous.

---

## Round 0.2 · Minor Revisions

After a few revision requests from one of the reviewers, your manuscript will be ready for acceptance.

Reviewer 1 ·

Basic reporting

I have received appropriate responses to my questions and generally understand them. However, there are some points I would like you to reconsider.

In the previous review, I requested additional explanations in the abstract for the reason why medicinal plants were utilized in this study. However, this has not yet been sufficiently addressed. Since this has been appropriately elaborated in the Introduction, please add a concise summary of the explanations provided in the Introduction to the abstract.

Line 9
In the Word file: adabtibility → adaptability
In the PDF file: Corrected to adaptability

Experimental design

no comment

Validity of the findings

no comment

Additional comments

I chose major revisions as my recommendation, but it is close to minor revisions.

Reviewer 3 ·

Basic reporting

Clear

Experimental design

Well defined

Validity of the findings

Relevant

Additional comments

Author attended to earlier comments

---

## Round 0.3 · accepted · Accept

I see that you have completed all the changes for acceptance for publication. Congratulations